# The acceptability of minimally invasive tissue sampling for cause of death determination in rural South Africa: A qualitative analysis

Laura-Lynne Brandt[1]*, Gift Mathebula[1], Zokwane Lucky Mondlane[1], Sara Jewett[2], Kathleen Kahn[1], Ryan Gregory Wagner[1], Jessica Price[1]

**1** SAMRC/Wits Rural Public Health and Health Transitions Research Unit (Agincourt), School of Public Health, Faculty of Health Sciences, University of the Witwatersrand, Johannesburg, South Africa, **2** Health & Society Division, School of Public Health, University of the Witwatersrand, Johannesburg, South Africa

* 2207756@students.wits.ac.za

## Abstract

### Background

Minimally invasive tissue sampling (MITS) has been used to determine cause of death in various low- and middle-income countries. However, information on the acceptability of this procedure in community-based deaths is limited; most studies have focused on facility-based deaths among children. This qualitative study describes factors affecting the prospective acceptability of MITS for community deaths across all ages in a rural South African community and reviews the utility of the Theoretical Framework for Acceptability (TFA).

### Methods

This qualitative study was conducted in the rural Agincourt Health and socio-Demographic Surveillance System (HDSS) site in Mpumalanga, South Africa. Thematic analysis was conducted on 20 in-depth interviews with residents who had experienced a death within the previous 2 years and 6 focus group discussions (FGDs) with key stakeholders. FGD groups included community members, healthcare workers, traditional healers, religious leaders and mortuary workers.

### Results

MITS was considered acceptable by interviewees, who posited that bereaved families had a strong desire to know cause of death, which would drive participation. Limited manipulation of the body and minimal disruption of burial practices were conditions that would make MITS more readily acceptable. Facilitators of participation included engaging with local traditional leaders, rigorous community education, transparency and openness regarding MITS activities, and the provision of emotional and psychological support to the bereaved. Whilst local beliefs did not forbid participation

which permits unrestricted use, distribution, and reproduction in any medium, provided the original author and source are credited.

**Data availability statement:** The full anonymized transcripts are not publicly available due to the sensitive nature of the data and the risk of participant identification. Excerpts of the transcripts relevant to the study have been included within the manuscript. These restrictions were imposed by the University of the Witwatersrand Human Research Ethics Committee (HREC Medical) and the Mpumalanga Provincial Health Research and Ethics Committee (MPHREC). Researchers who meet the criteria for access to confidential data may request the transcripts by contacting the corresponding author or the MPHREC at the following contact details: Email: floidy.wafawa-naka@wits.ac.za Tel:+27 (13) 766 3429.

**Funding:** This project has been funded by the MITS Alliance through funding from the Bill & Melinda Gates Foundation (INV-034017). The funding was awarded to the PI, KK, and team. The MRC/Wits Rural Public Health and Health Transitions Research Unit and Agincourt Health and Socio-Demographic Surveillance System, a node of the South African Population Research Infrastructure Network (SAPRIN), is supported by the Department of Science, Technology and Innovation, the University of the Witwatersrand, and the Medical Research Council, South Africa, and previously the Wellcome Trust, UK (grants 058893/Z/99/A; 069683/Z/02/Z; 085477/Z/08/Z; 085477/B/08/Z) Bill and Melinda Gates Foundation: https://www.gates-foundation.org/ MITS Alliance: https://mitsal-liance.org/ Department of Science, Technology and Innovation: https://www.dsti.gov.za/ The University of the Witwatersrand: https://www.wits.ac.za/ Medical Research Council, South Africa: https://www.samrc.ac.za/ Wellcome Trust, UK: https://wellcome.org/ The funders had no role in study design, data collection and analysis, decision to publish, or preparation of the manuscript.

**Competing interests:** The authors have declared that there are no competing interests.

in MITS, acceptability was limited for deaths in traditional healers and infants as it would disrupt burial in these groups. Rumours of organ-trafficking during autopsies made some participants wary of the MITS and a lack of trust in the research team could discourage participation.

## Conclusion

In Agincourt, MITS is an acceptable procedure among community members that are interested in knowing cause of death. Thorough community engagement, open communication and an empathetic approach to bereaved families are crucial for building community support for the implementation of MITS. The TFA provides a valuable outline for the assessment of acceptability but failed to account for trust dynamics between providers and participants. We propose the modification of the TFA to include the domain "trust in providers".

## Introduction

Despite the value of detailed cause of death information in managing health system resources only 42% of all deaths globally are registered with an informative cause of death recorded [1]. In South Africa, whilst 96% of deaths were registered in 2017, only 63% of registered deaths had a medically certified cause of death recorded [2]. In rural areas, the paucity of cause of death data is often even more pronounced as many deaths occur outside of health-facilities [3]. The lack of reliable cause of death data for community deaths and deaths from more remote areas result in the exclusion of these deaths from key health policy discussions, which can result in public health issues going unnoticed and unaddressed.

Complete diagnostic autopsy (CDA) is the "gold standard" method of investigating causes of death but there has been a decline in autopsy rates over the years [4–7]. Families often refuse to consent to CDA because it contradicts with their religious or cultural beliefs, they view it as unnecessary, or they have concerns about tissue retention by the institutions performing the autopsies [8,9]. In low-and middle-income countries (LMICs), the lack of capacity for CDA in combination with the proportion of deaths occurring outside healthcare facilities (and so without option of medical record review for certification of cause of death further confound the production of accurate cause of death data [10,11].

Minimally invasive tissue sampling (MITS) is a method of cause of death investigation that can be implemented in settings where CDA is not practical nor acceptable. During this procedure, tissue samples are taken via needle biopsy from targeted organs, in addition to blood and cerebrospinal fluid samples. These samples undergo microbiological, histological and molecular testing, and the results are then interpreted to ascertain cause of death [12]. MITS is less invasive, less costly and quicker to conduct that a CDA [13]. Studies from Mozambique, Switzerland and the Netherlands have shown that MITS has high concordance with CDA for cause of death [14–16].

The acceptability of an intervention is an important factor to consider when designing and planning interventions. As seen with CDA, a lack of acceptability among recipients significantly reduces the uptake of the intervention [17]. Work from sub-Saharan Africa and South Asia have shown that MITS is broadly acceptable and has described several factors effecting acceptability, including religious and cultural beliefs, burial practices and their understanding of the procedure [17–22]. However, the majority of MITS work has been restricted to facility-based deaths, deaths in children under 5 and stillbirths [17–24]. There is a need to understand the acceptability of MITS in community deaths across all ages in a rural setting, in order to inform the implementation of MITS in out-of-hospital, resource-limited settings.

The theoretical framework for acceptability (TFA), was developed to guide the assessment of the prospective, concurrent and retrospective acceptability of healthcare interventions (Fig. 1) [25]. Using or adapting existing conceptual frameworks to assess acceptability of an intervention enhances comparability with other studies and adds rigour to methods utilized to collect and analyse data [26]. To date, none of the studies on acceptability of MITS have described the use of the TFA or any alternative framework to map the findings in their context. Therefore, the utility of this framework for describing the acceptability of MITS is unknown.

This study is part of formative work conducted prior to the implementation of MITS in a rural community in South Africa. In this paper, we describe the prospective acceptability of MITS for cause of death determination in community deaths across all ages, identify key barriers and facilitators of participation, and assess the utility of the TFA in exploring the acceptability of MITS.

## Methods

This qualitative study consisted of in-depth interviews (IDIs) and focus groups discussions (FGDs) with community members and key stakeholders. The research team drew on phenomenological and broad ethnographic approaches to explore

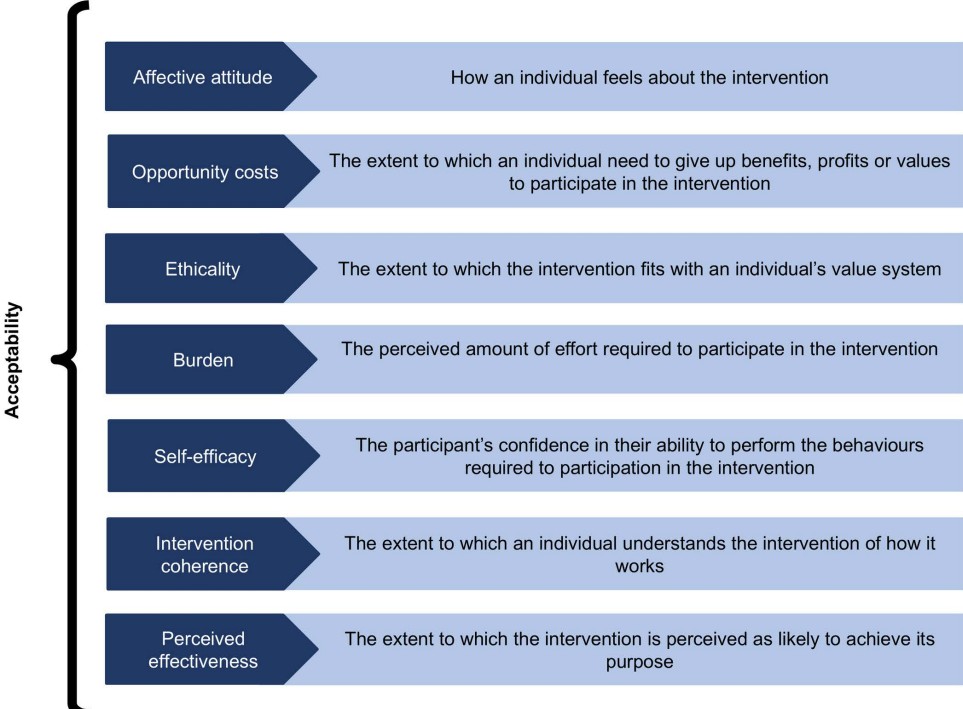

**Fig 1. The theoretical framework for acceptability (TFA) [25].**

beliefs and customs relating to death, perceptions of MITS and grieving practices in the community. This study is part of formative research conducted to better inform the implementation of MITS this specific context. The proposed study will aim to conduct MITS on deaths occurring outside of healthcare facilities, collecting samples of brain, lung, liver, heart, kidney and splenic tissue, in addition to blood and cerebrospinal fluid to determine cause of death [27,28].

## Study setting

The research was conducted in the Agincourt Health and socio-Demographic Surveillance System (HDSS) study area, in Bushbuckridge, Mpumalanga in north-eastern South Africa (Fig 2). The site is about 40 km from the Mozambique border and a third of the population consists of Mozambican immigrants [29]. The Agincourt HDSS was established in 1992 and is run by the South African Medical Research Council/Wits University Rural Public Health and Health Transitions Research Unit. There is thus a decades-long collaborative relationship with the occupants of the study area. The HDSS covers 31 villages and a population of some 117,000 people. Since its inception, all deaths in the study area are recorded and a verbal autopsy (VA) completed to determine probable cause of death [30,31]. The infrastructure in the area is poor; the few roads that are tarred are poorly maintained, a limited number of people have access to piped water and the supply to these pipes are inconsistent; though most homes have access to electricity, families are not always able to afford it. The

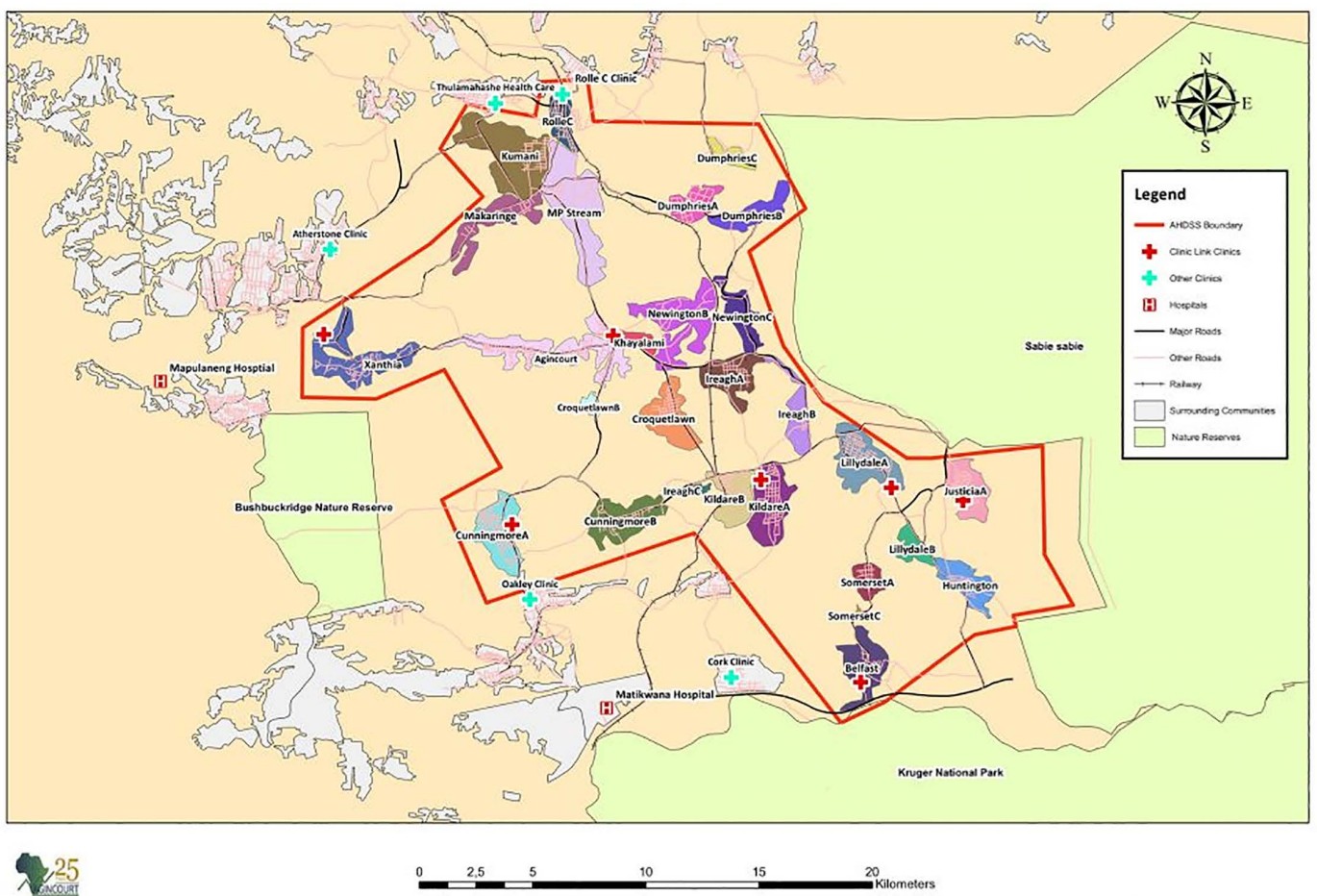

**Fig 2. The SAMRC/Wits Agincourt Research Unit study area [32].**

unemployment rate is extremely high and most households are reliant on government social grants (particularly the older persons grant and child support grant) as sources of income [29]. Healthcare services are accessible via one community health centre, 6 clinics and 3 nearby district hospitals 25–60 km away. Many community members also consult traditional healers when experiencing ill-health [30].

## Eligibility criteria

Individuals had to be over the age of 18 years and resident within the HDSS area. IDI participants must have participated in a verbal autopsy interview within the previous 24 months. FGD participants had to identify with one of the following groups: religious leaders, mortuary workers, healthcare workers, traditional leaders, traditional healers or a community member.

## Sampling

A purposive sampling strategy was used to identify participants for the IDIs. The Agincourt HDSS database was used to identify individuals who had experienced a death in the 2 years prior; participants could draw on the experience of a recent loss to formulate responses. Sample size for IDIs was determined when data saturation had occurred. We employed a combination of nomination and snowball sampling to sample participants for the FGDs. The Community Advisory Board (CAB) and the Agincourt Public Engagement office nominated community members that could take part in the FGDs according to key stakeholder groups (community members, traditional leaders, traditional healers, mortuary workers, healthcare workers or religious leaders); additional participants were identified through snowballing. All participants for both the IDIs and the FGDs had to be over the age of 18.

## Data collection

Socio-demographic information was recorded on the transcripts. The IDIs and FGDs were conducted using interview guides, by a team of three interviewers who were residents of the study area, spoke the local language (Xitsonga) and had experience doing fieldwork, including qualitative interviews. One of the co-authors, GM, who is from the study area, was part of the interview team and was present for all IDIs and FDGs. The interview guides for the IDIs and FDGs covered similar topics; general practices in community-based deaths, beliefs around deaths and perceptions of MITS for cause of death determination. The IDIs focused on assessing acceptability at the level of the individual and their family, whereas the FGDs aimed to provide the perspective of the community and to validate the findings of the IDIs. The interviewers conducted a short demonstration of how the MITS procedure would be conducted, including the needle that would be used. The interviews were conducted face-to-face, in Xitsonga.

For the IDIs, two interviewers approached potential participants at their homes to obtain consent to conduct and audio-record the interview. Some participants had a family member present during the interviews who assisted in answering questions at times. For the FGDs, two interviewers invited the participants to the research offices in Agincourt to conduct the discussion.

The interviewers took detailed field notes in English during the IDIs and FGDs to supplement the interviews and to support feedback to the larger research team that met during data collection. The interviewers audio recorded, de-identified, translated and transcribed the interviews and FGDs as they completed them, which were shared with the larger research team in order to discuss potential areas for further probing.

## Data analysis

We conducted thematic analysis, drawing on a hybrid-approach developed by Fereday and Muir [33]. The team used MAXQDA (VERBI Software, 2024) to manage the analysis process. Data collection and analysis was a collaborative and

iterative process; the interviewers and the research team met throughout data-collection and analysis to review transcripts and coding choices. The transcripts were dual-coded, using an inductive and deductive approach by a team of five coders (LB, GM, ZLM, SJ and JP). GM and ZLM were based at the research centre within the community, whilst the rest of the team worked remotely. GM, being from the study area, provided a deeper understanding of the local cultural context. The preliminary codebook was developed after the review of four interviews and was updated as new codes were identified in subsequent interviews. Inter-coder agreement was assessed, and major discrepancies were resolved with consensus meetings following which transcripts were merged to reflect the final coding decisions. LB constructed the themes from the codes. Themes were grouped under topics inspired by existing literature, for ease of comparison with other studies.

## Ethical considerations

Ethical approval for this study was obtained from the University of the Witwatersrand Human Research Ethics Committee (Medical) (Clearance Certificate no. M221192) and from the Mpumalanga Provincial Health Research and Ethics Committee (Clearance certificate no. MP_202303_009). Participation in the IDIs and FGDs was voluntary; the interviewers obtained written, informed consent prior to conducting interviews or discussions.

## Results

A total of 20 IDIs and 6 FDGs were completed (Table 1). The socio-demographic characteristics of the IDI participants are listed in Table 2. The median age was 55 years, and most of our participants were female (14/20; 70%). More than half (11/20; 55%) of the participants were pensioners and the vast majority (17/20; 85%) were Christian, with some specifying which church to which they belonged. A summary of the FGD participants is provided in Table 3.

### Findings

Themes were grouped into 3 topics from the data: 1.) benefits/advantages of MITS, 2.) anticipated barriers to participation and 3.) potential facilitators of participation. Table 4 outlines the topics and their associated themes identified from the transcripts.

### 1. Benefits/advantages of MITS

To community members that viewed MITS as a positive innovation, the procedure would be valuable to grieving families and the general community as it would provide cause of death, is minimally invasive and does not interfere with burial.

**Table 1. Target population and data-collection methods.**

| Population | Data-collection tool | No. of participants |
|---|---|---|
| Local community members who experienced the death of a family member in the last 2 years | IDIs | 20 |
| Traditional healers | FGD | 8 |
| Religious leaders | FGD | 8 |
| Funeral home/mortuary attendants | FGD | 10 |
| Healthcare workers | FGD | 8 |
| General community members group 1 | FGD | 8 |
| General community members group 2 | FGD | 9 |
| **Total** | | **71** |

**Table 2. Socio-demographic characteristics of IDI participants.**

| Characteristics | Total (n = 20) |
|---|---|
| Age (Median, Range) | 55.35 (20-86) |
| Sex (M: F) | 6:14 |
| Marital status n, (%) | |
| Single | 7 (35) |
| Married | 4 (20) |
| Separated | 2 (10) |
| Widowed | 7 (35) |
| Education n, (%) | |
| No formal education | 6 (30) |
| Primary education | 3 (15) |
| Secondary Education/Higher | 11 (55) |
| Employment n, (%) | |
| Unemployed | 5 (25) |
| Student | 1 (5) |
| Pensioner | 11 (55) |
| Waitress | 1 (5) |
| Security | 2 (10) |
| Religion n, (%) | |
| None | 3 (15) |
| Christian | 8 (40) |
| Zion Apostle<br>ZCC | 2 (10)<br>5 (25) |
| Nazarene | 2 (10) |
| Mean interview duration (mm:ss) | 56:06 |

**Table 3. Summary of FGD participants.**

| FGD group | Number of participants (n = 51) | Age (median, range) | Gender (Male:Female) | Duration (hh:mm:ss) |
|---|---|---|---|---|
| Mortuary workers | 10 | 38.5 (30 –45) | 5:5 | 01:25:40 |
| Healthcare workers | 7 | 53 (35-61) | 1:6 | 02:29:41 |
| Religious leaders | 8 | 59 (50-67) | 7:1 | 02:44:36 |
| Traditional healers | 8 | 52 (29-66) | 1:7 | 03:13:30 |
| Community members- Group 1 | 8 | 50.5 (32-69) | 4:4 | 02:38:49 |
| Community members- Group 2 | 9 | 51.7 (38-66) | 3:6 | 02:46:05 |

**Provides cause of death.** The fact that MITS would produce a cause of death was identified as a potential driver of acceptability. As one interviewee simply put it, '*"...I want to know about what took his life."* – (female, 20, IDI)

For those wanting to know cause of death, reasons varied. For some, understanding what killed a loved one and why they died was deemed crucial to healing and closure: *"I need closure, when I know why a person died, I will have closure."*- (female, 53, IDI)

Others pointed out how cause of death might help them identify preventative measures within their family systems to avoid further death and disease.

**Table 4. Topics and themes regarding the acceptability of MITS.**

| Topics | Themes |
| --- | --- |
| Benefits/advantages of MITS | Provides cause of death |
| | Minimally invasive |
| | No disruption of burial rituals in most cases |
| Anticipated barriers to participation | MITS disrupts the burial of infants and traditional healers |
| | Lack of trust in researchers and collaborating bodies |
| | Psychological strain |
| | Potential harms of MITS |
| Potential facilitators of participation | Rigorous community engagement |
| | Engagement of local stakeholders |
| | Transparency regarding the procedure and results |
| | Grief counselling and support |

*"That will help you know what killed them so that you can be careful because you will know that it kills."* – (female, 68, IDI)

Similarly, healthcare workers felt cause of death information could assist them with managing and educating their patients. Another motivation for accurate cause of death results was to resolve conflict and accusations of murder and witchcraft. According to local beliefs, witchcraft was a plausible cause of death among community members. Unfortunately, suspicions of witchcraft also lead to accusations and conflict within families and among community members. Participants felt that MITS could disprove cause of death due to witchcraft, allowing accused community members to be acquitted.

*"It is important to know what happened because once someone dies and the body is still in the house, families start to point fingers at each other and accuse each other of witchcraft. They need to know so that they will be able to deal with it properly."* –(female, age unknown, Traditional healers FGD)

Conversely, a few community members felt that knowing cause of death was not important, as explained by one community member: *"Ah I think it is the same. Even if they can research this it won't bring me back to life (laughing out loud)."* – (female, 40, IDI)

**Minimally invasive.** Community members believed that the body must be kept as intact as possible for burial, as intentionally compromising the remains would stop the spirit from finding peace. Researchers explained that the MITS procedures minimized disfigurement; the only remaining signs of the procedure would be needle-pricks and, in some cases, minimal bleeding. This lack of disfigurement, in addition to the small size of the tissue sample (a tissue core in a needle), contributed to the acceptability of the procedure, and made it preferable to conventional autopsy, which was seen as invasive and gruesome:

*"…is that they take a very small tissue in a very small area and the deceased doesn't feel pain anymore because he is already dead. Rather than taking a grinder to split the body of the deceased, that is a little bit harder, but this needle will take up less space, there won't be any bleeding, and we won't see any blood coming."* - (male, 69, General community members FDG)

**No disruption of burial and rituals.** The researchers explained that MITS needs to be conducted within 24 hours of death, and the sampling procedure itself can take about an hour. This was not a cause for concern, as most burials are conducted about a week after death. The short time required for MITS to be completed also allayed concerns about rituals. One FGD member explained:

> *"I don't think MITS will disturb anything because you don't take long. Families can do their cultural or religious practices while you do your work and leave."* – (female, 58, General community members FGD)

There were no religious or cultural beliefs that forbade conducting MITS procedures on the deceased. Community members (Christian, traditional or a combination of the two), believed that after death, the spirit separates from the body, so the spirit would not feel any pain from the MITS procedure.

2. Anticipated barriers to participation

**MITS disrupts burials in infants and traditional healers.** While MITS did not interfere with most types of burial, it would interfere with the timing of burial in infants (under the age of 1) and traditional healers. Whilst participants did not mention any objections to conducting MITS in these groups, these deaths are accompanied with special beliefs, rituals and burial practices, which would make tissue sampling logistically challenging.

Both infants and traditional healers are often buried within a day of death, either in the early morning or late afternoon.

> *"…if a child dies today chances are that it will be buried today. There are not much burial preparations that are being made to bury the child."* – (female, 32, IDI)

Furthermore, in some cases, the bodies are not taken to a mortuary. In the case of infants, they are often buried in the back yard of the family property and the burial is accompanied by strict and specific rituals which could limit participation. These burials are private affairs; only the mother of the child and a few female elders are allowed to carry out and witness the burial; no other community members attend. The perceived consequences for failing to perform burials as prescribed are severe, such as infertility in the mother of the deceased child.

> *"…when burying an infant, they ensure that the grave is not deeper than 1 metre when they dig, it's just a shallow, they are doing this to ensure that the woman who lost a child, conceive other babies in future."* – (male, 63, Religious leaders FGD)

Traditional healers also do not want their bodies to be taken to the mortuary when they die, preferring that their body is prepared for burial at home.

**Lack of trust in providers and collaborating bodies.** Community members might choose not to participate because they do not trust the research unit. A major concern linked to mistrust of the research unit was the issue of organ removal. Although removing whole organs or large pieces of tissue via the MITS procedure is not possible, there were some concerns that the project would be an excuse to gain access to the bodies for organ-trafficking.

> *"My concern will be that it might happen that when you take him, you will remove some of his body parts and he will be back incomplete, something like that."* – (female, 32, IDI)

Exploitation was also a concern. Some community members were worried that the research unit would take advantage of bereaved families to extract information and then neglect to respond to any significant distress the participants may be experiencing. These concerns were partly driven by experiences of ongoing verbal autopsy work in the study area.

*"They came and asked me those questions [the VA interview] that they wanted to ask me, then after that, they left, and I was not happy about how they did because I felt like I have given them the information that they were looking for but to me, they did nothing to help me."* – (female, 32, IDI)

The perceptions community members have of any parties linked to the SAMRC/Wits Agincourt Research Unit, including clinics and mortuaries, could affect acceptability.

**Psychological strain.** Some participants felt there would be a significant psychological burden associated with participating in MITS. Some community members expressed concern that they might not have the strength to discuss the death with the research team.

*"I would be unable to do anything (helpless) due to thinking and crying for my child or grandchild who died."* – (female, 86, IDI)

**Potential harms of MITS.** A few participants raised concerns regarding the sharing of cause of death information, as broadly sharing cause of death results could have unintended effects. One was regarding breaches in confidentiality. For example, disclosing that a deceased family member passed away from HIV-related causes could disclose the status of spouses or other community members by proxy. This could lead to stigmatization and conflict.

*"…he was diagnosed with HIV, but his wife is still alive, don't forget that his wife is not ours [a direct relative], I will go and tell people that she got AIDS too."* – (female, 58, General community members FDG 2)

Another concern was fear of litigation and damage to reputation among healthcare workers and mortuary attendants. They were concerned that cause of death results from MITS would contradict their own assessments or suggest a fault on their part, resulting in clients or patients taking legal action.

3. Potential facilitators of participation

**Rigorous community engagement.** Some community members said that thorough dissemination of information was important to improve participation. Community meetings and handing out flyers were recommended as good strategies to ensure wide-spread information dissemination, particularly as rumours and misinformation could pose a significant barrier to participation.

*"The project should be well known to the community that when someone dies, immediately, MITS should come first. I don't think there will be a challenge once we get to understand the information."* – (female, 66, Traditional healers FGD)

Beyond correcting misinformation, such engagement would provide practical information to community members about the process, such as who to contact.

**Engage local stakeholders.** Engaging local leaders, especially traditional leaders (*indunas*) was another important strategy to secure participation; gaining the approval of traditional leaders is important for gaining the trust of the community. Furthermore, traditional leaders act as channels of communication between the research unit and the community; the traditional leaders can call community meetings for research team members to present the project.

*"…if you can go to the induna (traditional leader) and explain everything thoroughly, the induna will invite community meeting saying that WITS people visited, and they want to educate us today. When people come, you get in and present/or educate us, no matter what!"* – (female, 60, IDI)

**Transparency regarding the procedure and results.** Participants counselled that the research team will need to maintain open communication with the family if they agree to engage with MITS. Potential participants must receive a clear, detailed description of what the procedure entails.

 

*"Don't hide anything, explain the procedure, and demonstrate the needle, once they get to understand the difference between the two. Those who know the real post-mortem will opt for the needles."* – (male, 50, Religious leaders FGD)

Another suggestion was that allowing family to accompany the body to the mortuary to verify that no organs were removed after the procedure would address any mistrust.

**Grief counselling and support.** Provision of adequate support by the research team is crucial to mitigate the psychological strain of participating in MITS. Fieldworkers must approach families in an empathetic manner, and it would be important to provide formal counselling or referral letters to family members that are severely distressed.

*"Things like counselling, in most cases, we don't have. It's rare to find a person going for counselling after someone has died; we just deal with grief on our own, until it passes, even if it hurts us painfully."* – (female, 33, IDI)

Emotional support from family members and the community is also important to facilitate participation.

*"I won't even have the powers to go I know myself. If my husband is there, he is the one who will accompany you to where you have chosen to perform the tests, then after you are done that's when the body will be taken to [mortuary] where we have joined."* – (female, 32, IDI)

Participants also explained that support could be shown by means of financial assistance or gifts. In the community it is customary that the whole community would assist with funeral preparations in some way, either through chores or through donations, as a way of expressing condolences and showing support.

*"... the community members are collecting the R10. There are also boys/men who will assist in digging the hole in preparation for the burial. Then there are those who will have prayer services on... we have those helping with their society schemes; we call it "muganga" [where women in the community will gather according to their parts of the community to assist with chores in the bereaved family] and some bringing firewood..."* – (male, 45, IDI)

The research team could employ similar strategies to show gratitude to participating families and to help mitigate the burden of participating.

*"...maybe when you come you will come with water or cold drink (soft drink) to say here please drink my sister so that you can also feel better."* – (female, 40, IDI)

## Discussion

This study describes the perceptions of community members and key stakeholders in Agincourt, South Africa regarding the proposed implementation of MITS for community deaths. These findings show that while several features of MITS are acceptable, there are various factors that researchers and implementers must consider to show respect to the community and to support participation in MITS. Importantly, respecting local beliefs and norms, thorough community engagement and a positive relationship with community members will facilitate acceptability and help inform project design.

Community members in Agincourt had a positive attitude towards MITS and expressed a willingness to participate, which is similar to findings from various settings in Africa and Asia [17,18,20,21,24,34]. This spoke to the "affective attitude" domain of the TFA, which is defined as the feelings an individual has about an intervention. A strong desire to know cause of death was a primary motivator for acceptance of MITS. Notably, we found that the desire for closure was an important reason for wanting to know cause of death; similar to the findings from a study that was conducted in Pakistan [18,24]. Participants recognised that cause of death data could be used to prevent further disease and death among

surviving community members. For example, understanding why an infant died could assist families in protecting future children [17,18,20].

Consistent with studies from Sub-Saharan Africa, witchcraft was frequently implicated as a cause of death by community members [17,18]. The desire to absolve community members from accusations of witchcraft also drove the desire to know cause of death. Furthermore, participants posited that suspicions of witchcraft could also distract from the true cause of death, therefore, impeding the family and community's ability to respond appropriately to health issues [35].

Similar to findings from other studies, local cultural and religious beliefs govern the access, preparation and burial of deceased bodies, which in turn influence the acceptability of post-mortem investigations in Agincourt [17–19]. This finding is described by the TFA domain of ethicality. Community members in Agincourt, believed that MITS would not cause any harm to the deceased's spirit. Conversely, work done in Muslim communities for example, has highlighted that any alteration of the body is prohibited due to concerns that it may harm the spirit [17,21,36].

Consistent with findings from seven different sites, MITS acceptance was conditional on it not interfering with rituals and burial practices [37]. This is challenging in communities where burial is typically conducted within 24 hours of death, so participation in MITS would disrupt these practices [17,21,36,38]. In the TFA, this would be considered an opportunity cost. However, in the study setting, concerns about disruption of burial practices were minimal for deaths in adults where burials are typically a week after death. However, there was an overlap with the timing of MITS activities and the timing of burial of infants and traditional healers, which would reduce participation for these groups.

The envisioned emotional strain associated with participating in MITS was noted by multiple respondents, with some participants doubting their ability to take part in MITS activities themselves. In India, MITS participants requested that they be provided with psychological support during the study and that researchers must approach bereaved families in a sensitive manner [20,39]; we recorded the same requests. Providing psychological support to bereaved families could minimise the psychological burden associated with participation and facilitate participation. These findings speak to the TFA domains of burden (the effort required to participate in the MITS project) and self-efficacy.

As suggested by previous studies [23,34], MITS was preferred to CDA as it limited disfigurement of the body. However, consistent with the domain of intervention coherence of the TFA, it is important that communities understand the MITS procedures and how these differ from CDA, to appreciate that it is minimally invasive. During the IDIs and FDGs, participants were shown the biopsy needles that would be used to facilitate a clear understanding of the MITS procedure. These participants emphasized that community members are more likely to refuse the study if they don't understand it. Rigorous, culturally appropriate community-based engagement strategies are crucial to ensure individuals are aware of the project [17,37]. Consistent with a study conducted in Mozambique [17], active engagement with local community leaders was considered to be a crucial step in implementing MITS in Agincourt.

Of note, a lack of trust in the research team was identified as a barrier to participation. In some settings, including Agincourt, mistrust in health institutions and research entities are driven by rumours of organ-trafficking occurring during CDAs, which complicate implementation [17,18,40]. An advantage of MITS is that the nature of the procedure would make the removal of whole organs impossible; however, participants need to trust the research unit and understand the procedure to appreciate that nuance [23]. One study emphasized the difficulty of implementation in research-naïve communities, where rumours and mistrust of health facilities were prevalent [17]. Interestingly, our study suggests that challenges with trust may persist, even in communities that are frequently subject to research. To illustrate, the Agincourt research unit has been conducting population surveillance in the study area for the past 30 years. Despite this longstanding engagement, some community members still had concerns that the research team would take advantage of them. These findings underscore the importance of a target population's trust in intervention providers in ensuring an intervention is acceptable.

Though the TFA was useful in conceptualizing the hypothetical acceptability of MITS in Agincourt, we were unable to comment on the perceived effectiveness of MITS; a limitation of assessing acceptability with hypothetical scenarios. Mistrust has been identified as a barrier to participation in MITS in several settings, suggesting that trust in providers is an

important contributor to the acceptability of MITS [17,18,22,23,40]. The current framework fails to account for the contribution of the level of trust between the intervention provider and the community to the overall acceptability of an intervention. We recommend the addition of an eighth domain to the TFA, trust in providers, to expand the concept of acceptability.

### Strengths and limitations

While this study provides valuable insights into the acceptability of MITS in the Agincourt HDSS, there are several limitations which must be considered. Firstly, interviews were conducted in Xitsonga – the home language of respondents. While this promoted maximal engagement and understanding, some nuances may be lost in the process of translation of interviews into English. Additionally, transcripts were not returned to participants for comment, precluding the opportunity to clarify vague responses. Various data sources allowed for triangulation which would partially mitigate this. Secondly, interviewers were local community members – while this may have helped in putting respondents at ease, it could also influence the responses of community members who may feel unwilling to share personal information with someone within the community. Thirdly, this study considered only the prospective acceptability of MITS. This, paired with the fact that a very limited number of community members had experienced a CDA, meant participants had to respond to the hypothetical scenario of participating in MITS. Finally, the research team was unable to conduct a FGD with traditional leaders who were recognised as an important stakeholder group, as they refused to meet prior to obtaining their chief's approval.

### Conclusion

In Agincourt, MITS is an acceptable alternative to CDAs for determining cause of death in community deaths, on the condition that MITS activities do not interfere with local burial practices. Researchers must be cognisant of the psychological well-being of bereaved family members and take steps to mitigate any additional emotional strain caused by the project. Researchers also need to engage local leaders and relevant stakeholders to assist with community engagement and to gain the trust of the community.

The TFA is useful for describing the hypothetical acceptability of MITS and future acceptability studies can benefit from its guidance. However, the framework does not currently account for the contribution of the trust between the community and the intervention providers to the overall acceptability of an intervention. Therefore, in addition to adapting the TFA for their research topic, implementation scientists should consider the additional domain of "Trust in providers" for describing the acceptability of healthcare interventions.

Future work should investigate the concurrent and retrospective acceptability of MITS in community deaths to provide further insight into the perceived effectiveness of the procedure and any issues the research team or the community may not have anticipated prior to implementation.

### Acknowledgments

We would like to thank the participants and management of the Agincourt Health and Socio-Demographic Surveillance System for their contributions to the data collected during this study.

### Author contributions

**Conceptualization:** Laura-Lynne Brandt, Sara Jewett, Kathleen Kahn, Ryan Gregory Wagner, Jessica Price.

**Data curation:** Gift Mathebula, Kathleen Kahn, Ryan Gregory Wagner, Jessica Price.

**Formal analysis:** Laura-Lynne Brandt, Gift Mathebula, Zokwane Lucky Mondlane, Sara Jewett, Kathleen Kahn, Ryan Gregory Wagner, Jessica Price.

**Funding acquisition:** Kathleen Kahn, Ryan Gregory Wagner, Jessica Price.

**Investigation:** Gift Mathebula, Kathleen Kahn, Ryan Gregory Wagner, Jessica Price.

**Methodology:** Zokwane Lucky Mondlane, Sara Jewett, Kathleen Kahn, Ryan Gregory Wagner, Jessica Price.

**Project administration:** Gift Mathebula, Kathleen Kahn, Ryan Gregory Wagner, Jessica Price.

**Supervision:** Kathleen Kahn, Ryan Gregory Wagner, Jessica Price.

**Visualization:** Laura-Lynne Brandt.

**Writing – original draft:** Laura-Lynne Brandt, Zokwane Lucky Mondlane.

**Writing – review & editing:** Gift Mathebula, Sara Jewett, Kathleen Kahn, Ryan Gregory Wagner, Jessica Price.

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
