## [Decision Letter · Decision Letter 0]

2 May 2025

PONE-D-24-56472The acceptability of Minimally Invasive Tissue Sampling for cause of death determination in rural South Africa: a qualitative analysisPLOS ONE

Dear Dr. Brandt,

Thank you for submitting your manuscript to PLOS ONE. After careful consideration, we feel that it has merit but does not fully meet PLOS ONE’s publication criteria as it currently stands. Therefore, we invite you to submit a revised version of the manuscript that addresses the points raised during the review process.

If applicable, we recommend that you deposit your laboratory protocols in protocols.io to enhance the reproducibility of your results. Protocols.io assigns your protocol its own identifier (DOI) so that it can be cited independently in the future. For instructions see: https://journals.plos.org/plosone/s/submission-guidelines#loc-laboratory-protocols. Additionally, PLOS ONE offers an option for publishing peer-reviewed Lab Protocol articles, which describe protocols hosted on protocols.io. Read more information on sharing protocols at . Additionally, PLOS ONE offers an option for publishing peer-reviewed Lab Protocol articles, which describe protocols hosted on protocols.io. Read more information on sharing protocols at https://plos.org/protocols?utm_medium=editorial-email&utm_source=authorletters&utm_campaign=protocols..

We look forward to receiving your revised manuscript.

Kind regards,

Jayeshkumar Kanani

Guest Editor

PLOS ONE

https://journals.plos.org/plosone/s/file?id=ba62/PLOSOne_formatting_sample_title_authors_affiliations.pdf..

[This project has been funded by the MITS Alliance through funding from the Bill & Melinda Gates Foundation (INV-034017). The funding was awarded to the PI, KK, and team. The MRC/Wits Rural Public Health and Health Transitions Research Unit and Agincourt Health and Socio-Demographic Surveillance System, a node of the South African Population Research Infrastructure Network (SAPRIN), is supported by the Department of Science, Technology and Innovation, the University of the Witwatersrand, and the Medical Research Council, South Africa, and previously the Wellcome Trust, UK (grants 058893/Z/99/A; 069683/Z/02/Z; 085477/Z/08/Z; 085477/B/08/Z)

Bill and Melinda Gates Foundation: https://www.gatesfoundation.org/

MITS Alliance: https://mitsalliance.org/

Department of Science, Technology and Innovation: https://www.dsti.gov.za/

The University of the Witwatersrand: https://www.wits.ac.za/

Medical Research Council, South Africa: https://www.samrc.ac.za/

Wellcome Trust, UK: https://wellcome.org/].

5. In the online submission form, you indicated that [Excerpts of the transcripts relevant to the study have been included within the manuscript. Given that the research was conducted with a small group of participants that may risk identification, any requests to access the full anonymised transcripts should be made directly to the corresponding author, with a motivation.].

Additional Editor Comments:

Dear author, please address the reviewers' suggestions.

Reviewers' comments:

Reviewer's Responses to Questions

**Comments to the Author**

1. Is the manuscript technically sound, and do the data support the conclusions?

Reviewer #1: Partly

Reviewer #2: Yes

Reviewer #3: Yes

2. Has the statistical analysis been performed appropriately and rigorously? 

Reviewer #1: Yes

Reviewer #2: Yes

Reviewer #3: Yes

3. Have the authors made all data underlying the findings in their manuscript fully available?

Reviewer #1: Yes

Reviewer #2: Yes

Reviewer #3: Yes

4. Is the manuscript presented in an intelligible fashion and written in standard English?

Reviewer #1: Yes

Reviewer #2: Yes

Reviewer #3: Yes

5. Review Comments to the Author

Reviewer #1: Following clarification needed.

For a scientific study which will be helpful to the community, why so remote area is selected?

The study is based only on the basis of question and answer by interview or any follow up study.

Line 28-29 experience of death in family, How it will help in study? does this include the experience of cause of death investigation and any disgusting feeling by them during that procedure.

Does inform consent was in camera? there is no clarification.

There is no clarification about the type of tissue and from which site of body samples will be collected.

Who will collect the samples and who will process them?

What will be the cost effectiveness of such investigations?

If the community agree on the study, does author has proceeded for sampling and if yes what was the results?

Is the technique is applicable only to nature undiagnosed deaths or it can be used for unnatural deaths to know cause of death? Have author conducted any such study by taking minimum tissues? If yes what was the result?

What is the total population of study area?

Is the tissue sampling is under sonography or CT guided? If yes then whether these facilities are available there? or the investigator have to transport the samples to other center.

Reviewer #2: The authors address the issue of acceptability of MITS for determining the cause of death in rural South Africa

The paper is well written and clear.

Recommendations: Accept

My Comments:

Introduction: Line 53 “ a useable” – can be replaced with a better word

Methodology: No change

Results: 197 Table 3. Summary of FGD participants – Table incomplete?

Conclusions: No change

Reviewer #3: It is an very interesting paper, that fulfill and regard many aspects in the autopsy procedure acceptance, even complete or minimally invasive, in low-income settings. The writing was very clear and sound, with a right lenght of text. The methods are good, and suitable with the strenghts and limitations discussion.

6. PLOS authors have the option to publish the peer review history of their article (what does this mean?). If published, this will include your full peer review and any attached files.). If published, this will include your full peer review and any attached files.

.

Reviewer #1: No

Reviewer #2: **Yes:** Mary MathewMary Mathew

Reviewer #3: No

While revising your submission, please upload your figure files to the Preflight Analysis and Conversion Engine (PACE) digital diagnostic tool, https://pacev2.apexcovantage.com/. PACE helps ensure that figures meet PLOS requirements. To use PACE, you must first register as a user. Registration is free. Then, login and navigate to the UPLOAD tab, where you will find detailed instructions on how to use the tool. If you encounter any issues or have any questions when using PACE, please email PLOS at . PACE helps ensure that figures meet PLOS requirements. To use PACE, you must first register as a user. Registration is free. Then, login and navigate to the UPLOAD tab, where you will find detailed instructions on how to use the tool. If you encounter any issues or have any questions when using PACE, please email PLOS at figures@plos.org. Please note that Supporting Information files do not need this step.. Please note that Supporting Information files do not need this step.

---

## [Author Response · Author response to Decision Letter 1]

1 Dec 2025

Dear Editors and Reviewers,

We wish to thank the reviewers and editors for their valuable feedback regarding the manuscript titled “The acceptability of minimally invasive tissue sampling for cause of death determination in rural South Africa: a qualitative analysis”. We greatly value the feedback provided and will summarise our response in this comment. Please see the documents titled, "Cover letter" and "Response to reviewers" for more detailed clarifications.

Reviewer 1:

For a scientific study which will be helpful to the community, why so remote area is selected?

- This study was purposefully set in a rural community as their are more gaps in cause of death data in these low-resource settings. Minimally invasive tissue sampling (MITS) could be a potentially be useful for filling these gaps. This study specifically aims to identify cultural and structural barriers to the implementation of MITS in these settings.

The study is based only on the basis of question and answer by interview or any follow up study.

- This study relies purely on in-depth interviews and focus group discussions as a source of data. A follow-up study is underway, however, the results of this study is not yet available.

Line 28-29 experience of death in family, How it will help in study? does this include the experience of cause of death investigation and any disgusting feeling by them during that procedure.

- Participant that have recently experienced a death would have more insight into the emotional distress and the ritual practices surrounding deaths in this setting. These participants would be better equipped to share their personal experiences and how amenable grieving families would be to participating in a MITS study. The vast majority of residents in this setting have no experience with autopsies, so this was not part of our inclusion criteria. We provided demonstrations on how the MITS procedure is performed to ensure their responses were well informed.

Does inform consent was in camera? there is no clarification.

- We recorded the interviews with audio recorders, not cameras. We apologise for not clarifying this in the initial manuscript. The updated manuscript includes which recording modalities was used and how consent was obtained.

There is no clarification about the type of tissue and from which site of body samples will be collected.

- We apologise for the lack of clarity. This study is a formative study performed prior to the start of the MITS intervention study, in order to better understand whether the intervention would be acceptable and to inform the design of the MITS intervention study. We included a section in the manuscript detailing the procedures undertaken during the MITS process, included what tissues will be sampled. Please see the "Response to reviewers" document and the updated manuscript for more details.

Who will collect the samples and who will process them?

- We did not include this level of detail regarding the methodology of the MITS intervention study, as this manuscript is not about the MITS intervention study. This was a qualitative study undertaken to assisting with the planning of the MITS intervention study to optimize implementation efforts. This paper specifically discusses the acceptability of MITS and explore cultural and structural barriers to its implementation.

What will be the cost effectiveness of such investigations?

- Unfortunately, no studies to date have investigated the cost-effectiveness of MITS and it is also beyond the scope of this study.

If the community agree on the study, does author has proceeded for sampling and if yes what was the results?

- The MITs intervention study has commenced (after ethical clearance and permission from the relevant authorities). This study is still ongoing and the result of this study will be published in a separate paper.

Is the technique is applicable only to nature undiagnosed deaths or it can be used for unnatural deaths to know cause of death? Have author conducted any such study by taking minimum tissues? If yes what was the result?

- The MITS technique is relatively novel and its application to unnatural deaths has not been investigated. This study is part of formative work conducted prior to the MITS intervention study and will only discuss the acceptability of the procedure. The MITS study is still ongoing at the result will be discussed in a different paper.

What is the total population of study area?

- The latest estimates included about 117 000 residents within the Agincourt Health and socioDemographic Surveillance system platform.

Is the tissue sampling is under sonography or CT guided? If yes then whether these facilities are available there? or the investigator have to transport the samples to other center.

- The research team is not utilizing any imaging modalities to guide the MITS process. The team is working on a protocol to incorporate portable ultrasound-guided sampling into the procedure.

Reviewer 2:

Introduction: Line 53 “ a useable” – can be replaced with a better word

- Replaced “a useable” with “an informative”

Methodology: No change

Results: 197 Table 3. Summary of FGD participants – Table incomplete?

- Our apologies if there was an error in uploading table 3. We have double checked that the table is not missing any information and have uploaded a new copy.

Conclusions: No change

Reviewer 3:

- No changes noted.

Below we will comment on the feedback for the editors regarding issues with our initial submission.

1. Financial disclosure revised. Please find the updated statement with the added statement regarding the Role of Funders below.

“This project has been funded by the MITS Surveillance Alliance through funding from the Gates Foundation (INV-034017). The funding was awarded to the principal investigator, KK, and team. The SAMRC/Wits Rural Public Health and Health Transitions Research Unit and Agincourt Health and socio-Demographic Surveillance System, a node of the South African Population Research Infrastructure Network (SAPRIN), is supported by the Department of Science, Technology and Innovation, the University of the Witwatersrand, and the Medical Research Council, South Africa, and previously the Wellcome Trust, UK (grants 058893/Z/99/A; 069683/Z/02/Z; 085477/Z/08/Z; 085477/B/08/Z)

Gates Foundation: https://www.gatesfoundation.org/

MITS Surveillance Alliance: https://mitsalliance.org/

Department of Science, Technology and Innovation: https://www.dsti.gov.za/

The University of the Witwatersrand: https://www.wits.ac.za/

South African Medical Research Council, South Africa: https://www.samrc.ac.za/

2. Funding information section on the submission form has been updated to match financial disclosure as above.

3. Regarding data availability. We do, however, have concerns regarding making our dataset available to the public. We recognise that data sharing is important for transparency and replicability in research, however we are concerned that even a de-identified set of interview transcripts would risk breaking confidentiality with the participants included in this study. The interviews were conducted in a small community, easily identified because of its institutional base and research history. and Participants shared deeply personal and unique experiences, recounting events that led to deaths which would be recognised by other members of the community. To remove these elements from the transcripts in an attempt to de-identify the transcripts would remove core content which forms the basis of the thematic analysis included in this paper and would defeat the purpose of sharing the raw data. We hope that given this special circumstance we may be exempted from this data-sharing policy. We are happy to share the transcripts elements of the data with individual researchers on request, where data protection and privacy conditions can be assured. We would also be happy to upload our code book to the data repository if this would be helpful.

This concludes our comments on the responses from the editors and reviewers. We would like to thank them again for their valuable input.

---

## [Decision Letter · Decision Letter 1]

9 Dec 2025

PONE-D-24-56472R1The acceptability of Minimally Invasive Tissue Sampling for cause of death determination in rural South Africa: a qualitative analysisPLOS One

Dear Dr. Brandt,

Thank you for submitting your manuscript to PLOS ONE. After careful consideration, we feel that it has merit but does not fully meet PLOS ONE’s publication criteria as it currently stands. Therefore, we invite you to submit a revised version of the manuscript that addresses the points raised during the review process.

If applicable, we recommend that you deposit your laboratory protocols in protocols.io to enhance the reproducibility of your results. Protocols.io assigns your protocol its own identifier (DOI) so that it can be cited independently in the future. For instructions see: https://journals.plos.org/plosone/s/submission-guidelines#loc-laboratory-protocols. Additionally, PLOS ONE offers an option for publishing peer-reviewed Lab Protocol articles, which describe protocols hosted on protocols.io. Read more information on sharing protocols at . Additionally, PLOS ONE offers an option for publishing peer-reviewed Lab Protocol articles, which describe protocols hosted on protocols.io. Read more information on sharing protocols at https://plos.org/protocols?utm_medium=editorial-email&utm_source=authorletters&utm_campaign=protocols..

We look forward to receiving your revised manuscript.

Kind regards,

Jayeshkumar Kanani

Guest Editor

PLOS One

Journal Requirements:

Additional Editor Comments:

Reviewer 1: Still the point is not clear to me, why the author chose to study on the interview basis? Why general public was involved in a study which require authorization from higher authorities? The author has not discussed about cost effectiveness of the test because if it is free burdone to goventment and if chargeble then people have to know the cost per test.

Reviewers' comments:

Reviewer's Responses to Questions

**Comments to the Author**

1. If the authors have adequately addressed your comments raised in a previous round of review and you feel that this manuscript is now acceptable for publication, you may indicate that here to bypass the “Comments to the Author” section, enter your conflict of interest statement in the “Confidential to Editor” section, and submit your "Accept" recommendation.

Reviewer #1: All comments have been addressed

2. Is the manuscript technically sound, and do the data support the conclusions?

Reviewer #1: Partly

3. Has the statistical analysis been performed appropriately and rigorously? 

Reviewer #1: Yes

4. Have the authors made all data underlying the findings in their manuscript fully available?

Reviewer #1: Yes

5. Is the manuscript presented in an intelligible fashion and written in standard English?

Reviewer #1: No

6. Review Comments to the Author

Reviewer #1: Still the point is not clear to me, why the author chose to study on the interview basis? Why general public was involved in a study which require authorization from higher authorities? The author has not discussed about cost effectiveness of the test because if it is free burdone to goventment and if chargeble then people have to know the cost per test.

7. PLOS authors have the option to publish the peer review history of their article (what does this mean?). If published, this will include your full peer review and any attached files.). If published, this will include your full peer review and any attached files.

.

Reviewer #1: No

---

## [Author Response · Author response to Decision Letter 2]

25 Feb 2026

Dear Reviewers,

RE: MANUSCRIPT NO: PONE-D-24-56472.Title: “The acceptability of minimally invasive tissue sampling for cause of death determination in rural South Africa: a qualitative analysis”

We would like to thank the reviewers for considering our updated manuscript titled “The acceptability of minimally invasive tissue sampling for cause of death determination in rural South Africa: a qualitative analysis”. We understand that there are still some points that you found to be unclear and we are happy to provide further explanation.

The responses to the reviewer comments are as follows:

Still the point is not clear to me, why the author chose to study on the interview basis?

Previous studies exploring the acceptability of MITS also utilised interviews to collect their data. Our rationale for using interviews and focus group discussions to explore this topic was to get a detailed account of factors affecting the acceptability of MITS and to explore other practical aspects around the implementation of MITS in the Agincourt HDSS area. Quantitative methods would not have allowed us to get the depth of information we needed to inform the implementation of MITS in this setting. I have included references to other studies that have explored the acceptability of MITS using qualitative interviews below. These studies have also been cited in the manuscript.

Why general public was involved in a study which require authorization from higher authorities?

This study was approved by the University of the Witwatersrand Human Research Ethics Committee (Medical) and from the Mpumalanga Provincial Health Research and Ethics Committee as these were the local ethics committees responsible for enforcing ethical standards in research in the area. This was not a national project, but a research study conducted only within the Agincourt HDSS area.

The author has not discussed about cost effectiveness of the test because if it is free burdone to goventment and if chargeble then people have to know the cost per test.

This study is purely for research purposes, and all the costs of the project are covered by research grants. This is not a service that is provided by the South African government currently nor a service that is available to the general community on request at their own cost. For this study there is no cost to the consumer or to the government. A cost effectiveness analysis can be conducted at a later stage once full costing implications are better understood. The research presented in this manuscript was a qualitative investigation into the acceptability of MITS in a rural community and so costings were not reflective of the MITS procedure itself.

List of article investigating the acceptability of MITS:

1. Magaço A, Alonso Y, Maixenchs M, Ambrosio C, Sitoe A, Vitorino P, et al. A Qualitative Assessment of Community Acceptability and Its Determinants in the Implementation of Minimally Invasive Tissue Sampling in Children in Quelimane City, Mozambique. Am J Trop Med Hyg. 2023 Apr 10;tpmd220343.

2. Ngwenya N, Coplan D, Nzenze S, Myburgh N, Madhi S. Community acceptability of minimally invasive autopsy (MIA) in children under five years of age in Soweto, South Africa. Anthropol South Afr. 2017 June 2;40(2):108–21.

3. O’Mara Sage E, Munguambe KR, Blevins J, Guilaze R, Kosia B, Maixenchs M, et al. Investigating the Feasibility of Child Mortality Surveillance With Postmortem Tissue Sampling: Generating Constructs and Variables to Strengthen Validity and Reliability in Qualitative Research. Clin Infect Dis Off Publ Infect Dis Soc Am. 2019 Oct 9;69(Suppl 4):S291–301.

4. Das MK, Arora NK, Kaur G, Malik P, Kumari M, Joshi S, et al. Perceptions of family, community and religious leaders and acceptability for minimal invasive tissue sampling to identify the cause of death in under-five deaths and stillbirths in North India: a qualitative study. Reprod Health. 2021 Aug 4;18(1):168.

5. Feroz A, Ibrahim MN, McClure EM, Ali AS, Tikmani SS, Reza S, et al. Perceptions of parents and religious leaders regarding minimal invasive tissue sampling to identify the cause of death in stillbirths and neonates: Results from a qualitative study. Reprod Health [Internet]. 2019;16(1). Available from: https://www.scopus.com/inward/record.uri?eid=2-s2.0-85065569015&doi=10.1186%2fs12978-019-0730-9&partnerID=40&md5=330380b6971fd968805251084a64f4ed

6. Lawrence S, Namusanya D, Hamuza A, Huwa C, Chasweka D, Kelley M, et al. Hypothetical acceptability of hospital-based post-mortem pediatric minimally invasive tissue sampling in Malawi: The role of complex social relationships. PloS One. 2021;16(2):e0246369.

7. Rugwizangoga B, Niyibizi JB, Ndayisaba MC, Musoni E, Manirakiza F, Uwineza A, et al. Exploring Perceptions and Acceptance of Minimally Invasive Tissue Sampling among Bereaved Relatives and Health-Care Professionals in Rwanda. J Multidiscip Healthc. 2021 Dec 15;14:3421–7.

8. Ahsan N, Malik FA, Mughis W, Jabeen R, Mehboob S, Allana R, et al. Postmortem minimally invasive tissue sampling in communities: exploring perceptions of families, funeral workers, religious and community leaders and healthcare providers from Pakistan. BMC Health Serv Res. 2023 Dec 13;23(1):1402.

Thank you for considering this manuscript and giving me the opportunity to respond to your queries. I look forward to your response.

Kind regards,

Laura-Lynne Brandt

---

## [Editor Report · Decision Letter 2]

2 Mar 2026

The acceptability of Minimally Invasive Tissue Sampling for cause of death determination in rural South Africa: a qualitative analysis

PONE-D-24-56472R2

Dear Dr. Brandt,

We’re pleased to inform you that your manuscript has been judged scientifically suitable for publication and will be formally accepted for publication once it meets all outstanding technical requirements.

An invoice will be generated when your article is formally accepted. Please note, if your institution has a publishing partnership with PLOS and your article meets the relevant criteria, all or part of your publication costs will be covered. Please make sure your user information is up-to-date by logging into Editorial Manager at Editorial Manager® and clicking the ‘Update My Information' link at the top of the page. For questions related to billing, please contact  and clicking the ‘Update My Information' link at the top of the page. For questions related to billing, please contact billing support..

Kind regards,

Jayeshkumar Kanani

Guest Editor

PLOS One

Additional Editor Comments (optional):

The authors have carefully considered all reviewer comments and have provided appropriate revisions and clarifications throughout the manuscript. The revised version demonstrates significant improvement in structure, scientific rigor, and overall presentation. The concerns raised during the initial review have been satisfactorily resolved. Therefore, I believe the manuscript is now suitable and acceptable for publication.
---

## [Editor Report · Acceptance letter]

PONE-D-24-56472R2

PLOS One

Dear Dr. Brandt,

I'm pleased to inform you that your manuscript has been deemed suitable for publication in PLOS One. Congratulations! Your manuscript is now being handed over to our production team.

Kind regards,

on behalf of

Dr. Jayeshkumar Kanani

Guest Editor

PLOS One